# Climate and water-table levels regulate peat accumulation rates across Europe

Graeme T. Swindles[1,2]*, Donal J. Mullan[3], Neil T. Brannigan[4], Richard E. Fewster[1],
Thomas G. Sim[5], Angela Gallego-Sala[6], Maarten Blaauw[7], Mariusz Lamentowicz[8],
Vincent E.J. Jassey[9], Katarzyna Marcisz[8], Sophie M. Green[3], Thomas P. Roland[6],
Julie Loisel[10], Matthew J. Amesbury[11], Antony Blundell[12], Frank M. Chambers[13],
Dan J. Charman[14], Callum R.C. Evans[3], Angelica Feurdean[15,16], Jennifer M. Galloway[17],
Mariusz Gałka[18], Edgar Karofeld[19], Evelyn M. Keaveney[1], Atte Korhola[20],
Łukasz Lamentowicz[8], Peter Langdon[21], Dmitri Mauquoy[22], Michelle M. McKeown[23],
Edward A. D. Mitchell[24,25], Gill Plunkett[26], Helen M. Roe[3], T. Edward Turner[27],
Ülle Sillasoo[28], Minna Väliranta[29], Marjolein van der Linden[30], Barry Warner[31]

1 Geography and [14]Chrono Centre, School of Natural and Built Environment, Queen's University Belfast, Belfast, United Kingdom, 2 Ottawa-Carleton Geoscience Centre and Department of Earth Sciences, Carleton University, Ottawa, Ontario, Canada, 3 Geography, School of Natural and Built Environment, Queen's University Belfast, Belfast, United Kingdom, 4 School of Geography and Environmental Sciences, Ulster University, Coleraine, Northern Ireland, United Kingdom, 5 Forest Research, Northern Research Station, Roslin, Midlothian, United Kingdom, 6 Geography, Faculty of Environment, Science and Economy, University of Exeter, Exeter, United Kingdom, 7 Archaeology and Palaeoecology and [14]Chrono Centre, School of Natural and Built Environment, Queen's University Belfast, Belfast, United Kingdom, 8 Climate Change Ecology Research Unit, Faculty of Geographical and Geological Sciences, Adam Mickiewicz University, Poznań, Poland, 9 Université de Toulouse, Toulouse INP, CNRS, IRD, CRBE, Toulouse, France, 10 Department of Geography, University of Nevada in Reno, Reno, Nevada, United States of America, 11 Research Services, University of Exeter, Innovation Centre Phase 2, Exeter, United Kingdom, 12 School of Geography, University of Leeds, Leeds, United Kingdom, 13 Centre for Environmental Change and Quaternary Research, School of Natural and Social Sciences, University of Gloucestershire, Cheltenham, United Kingdom, 14 Geography, College of Life and Environmental Sciences, University of Exeter, Exeter, United Kingdom, 15 Goethe University, Institute of Physical Geography, Frankfurt am Main, Germany, 16 Department of Geology, Babeş-Bolyai University, Cluj-Napoca, Romania, 17 Geological Survey of Canada/Commission géologique du Canada, Calgary, Alberta, Canada, 18 Department of Biogeography, Paleoecology and Nature Conservation, Faculty of Biology and Environmental Protection, University of Lodz, Lodz, Poland, 19 Institute of Ecology and Earth Sciences, University of Tartu, Tartu, Estonia, 20 Environmental Change Research Unit, Faculty of Biological and Environmental Sciences, University of Helsinki, Helsinki, Finland, 21 School of Geography and Environmental Science, University of Southampton, Southampton, United Kingdom, 22 School of Geosciences, The University of Aberdeen, Aberdeen, United Kingdom, 23 Department of Geography and Environmental Research Institute, University College Cork, Cork, Ireland, 24 Laboratory of Soil Biodiversity, Institute of Biology, University of Neuchâtel, Neuchâtel, Switzerland, 25 Jardin Botanique de Neuchâtel, Neuchâtel, Switzerland, 26 Archaeology and Palaeoecology, School of Natural and Built Environment, Queen's University Belfast, Belfast, United Kingdom, 27 Forestry and Land Scotland, South Region, Dumfries, United Kingdom, 28 Gaia School in Tallinn, Tallinn, Estonia, 29 Environmental Change Research Unit, Ecosystems and environment research programme, University of Helsinki, Helsinki, Finland, 30 BIAX Consult, Zaandam, the Netherlands, 31 Department of Earth and Environmental Sciences, University of Waterloo, Waterloo, Ontario, Canada

* g.swindles@qub.ac.uk



**Data availability statement:** All relevant data are within the manuscript and its Supporting Information files.

**Funding:** We acknowledge all the organizations that have funded the data used in this analysis: Academy of Finland; Department for Employment and Learning (Northern Ireland); European Commission (Fifth Framework); INTERACT (European Community's Seventh Framework Programme); Irish Discovery Programme; Leverhulme Trust; National Science Centre (Poland); Natural Environment Research Council (UK); Natural Sciences and Engineering Research Council of Canada; Netherlands Organization for Scientific Research; Polish National Science Centre (2021/41/B/ST10/00060 and 2021/03/Y/ST10/00093 [BIODIVRESTORE]); Quaternary Research Association; Swiss Contribution to the enlarged European Union; Swiss Federal Office for Education and Science; Swiss National Science Foundation; World University Network; Wüthrich Fund (University of Neuchâtel); and Yorkshire Water. T.G.S. was funded by the Leeds–York Natural Environment Research Council (NERC) Doctoral Training Partnership (grant no. NE/L002574/1). T.E.T. acknowledges NERC Doctoral Training Grant no. NE/G52398X/1. JMGs contribution represents NRCan contribution number/Numéro de contribution de RNCan: 20230392. This paper is a contribution to the PAGES C-PEAT group. PAGES is supported by the Swiss Academy of Sciences and Future Earth. This research was supported by a grant to G.T.S from the UK Leverhulme Trust (Grant No. RPG-2021-354) and a grant to E.M.K from UKRI NERC (Grant No. UKRI182).

**Competing interests:** The authors have declared that no competing interests exist.

# Abstract

## Background

Peatlands are globally-important carbon sinks at risk of degradation from climate change and direct human impacts, including drainage and burning. Peat accumulates when there is a positive mass balance between plant productivity inputs and litter/peat decomposition losses. However, the factors influencing the rate of peat accumulation over time are still poorly understood.

## Methodology/Principal Findings

We examine apparent peat accumulation rates (aPAR) during the last two millennia from 28 well-dated, intact European peatlands and find a range of between 0.005 and 0.448 cm yr$^{-1}$ (mean = 0.118 cm yr$^{-1}$). Our work provides important context for the commonplace assertion that European peatlands accumulate at ~0.1 cm per year. The highest aPAR values are found in the Scandinavian and Baltic regions, in contrast to Britain, Ireland, and Continental Europe. We find that summer temperature is a significant climatic control on aPAR across our European sites. Furthermore, a significant relationship is observed between aPAR and water-table depth (reconstructed from testate-amoeba subfossils), suggesting that higher aPAR levels are often associated with wetter conditions. We also note that the highest values of aPAR are found when the water table is within 5–10 cm of the peatland surface. aPAR is generally low when water table depths are < 0 cm (standing water) or > 25 cm, which may relate to a decrease in plant productivity and increased decomposition losses, respectively. Model fitting indicates that the optimal water table depth (WTD) for maximum aPAR is ~10 cm.

## Conclusions/Significance

Our study suggests that, in some European peatlands, higher summer temperatures may enhance growth rates, but only if a sufficiently high water table is maintained. In addition, our findings corroborate contemporary observational and experimental studies that have suggested an average water-table depth of ~10 cm is optimal to enable rapid peat growth and therefore carbon sequestration in the long term. This has important implications for peatland restoration and rewetting strategies, in global efforts to mitigate climate change.

## Introduction

Peatlands comprise only ~3% of the global landmass but are globally important habitats, carbon (C) stores and valuable archives of past environmental and climatic change [1–3]. European peatlands contain approximately half of the continent's total soil organic carbon, and five times more carbon than its forests [4,5]. Peatlands

accumulate peat when plant inputs exceed peat losses through decomposition. Peat accumulation can therefore be considered a mass balance where net growth or loss is dependent on the difference between inputs and outputs of organic carbon [6,7]. Natural peatlands exhibit a near-surface water table, which slows down C losses from microbial catabolic activities and enables peat accumulation [8]. Vegetation composition is a strong control on peat accumulation rates, with *Sphagnum*-dominated bogs often the fastest peat accumulators [e.g., 9–11].

Many European peatlands have been damaged in recent centuries through human activities including burning, drainage, peat extraction, forestry, nutrient pollution and land-use changes [12]. In addition, climate change may have negative consequences for some peatland ecosystems as warmer temperatures and shifts in precipitation patterns may lead to deepening water tables, which could in turn lead to greater peat decomposition and loss of soil organic carbon [13], although climate change may also increase peat accumulation in certain regions [1,14].

Major efforts are underway to conserve and restore degraded peatlands across Europe (e.g., LIFE Peat Restore). The primary method for restoring peatlands involves "re-wetting" or re-establishing the natural flow of water and soil saturation to the peatland ecosystem, which may include blocking artificial drains or canals [15]. The anticipation is that drain blocking will lead to restoration of a healthy peatland ecosystem, and in time, the 'stabilised' peatland will begin to accumulate peat and/or act again as a long-term carbon store [16]. There is also evidence to suggest that damaged peatlands can self-repair, including the spontaneous recovery of degraded peatlands where an ecological regime shift from erosion to renewed carbon accumulation has been observed [17,18]. Peatlands can thus be considered complex adaptive systems, demonstrating the capacity to self-regulate in terms of vegetation and hydrological functioning [19,20].

However, there are many gaps in scientific understanding of the factors behind peat formation and what causes the peatland carbon sink capacity to destabilise and re-stabilise. If the key processes involved can be elucidated, researchers will be better placed to model how peatlands respond to ongoing climate changes and to advise how to manage peatland carbon stocks more effectively. Current efforts to restore peatlands are only partly based on scientific evidence and, without process-based understanding, it is difficult to predict the long-term outcome of management approaches – including the possibility that restoration may be ineffective in transforming peatlands into carbon sinks [see 21]. This study uses palaeoenvironmental data from 28 well-dated, intact European peat bogs (Fig 1) to determine the climatic and hydrological controls on the apparent rates of peat accumulation (aPAR) (Table 1, Fig 1). An understanding of how fast peatlands grow vertically (i.e., aPAR) may also have important implications for: i) robust determination of carbon (C) accumulation rates; ii) determining the temporal resolution of palaeoenvironmental profiles; and iii) validation of peatland development models.

## Results

Peat-core chronologies were determined using Bayesian age models (S1) and used to calculate temporal changes in aPAR at a 1-cm resolution for each core. Trends in aPAR data were then analysed against hydrological (water-table depth) and climatic (gridded reanalysis) datasets. We find that aPAR in our European sites range from 0.005 to 0.448 cm yr$^{-1}$ (Fig 2) and the average of the inter-site means ($n = 28$) is 0.118 cm yr$^{-1}$ (average of all sample depths ($n = 1732$) is 0.140 cm yr$^{-1}$).

To reveal long-term trends that reflect the typical functioning of each site, we consider the site-based aPAR averages, maxima and minima for further statistical analysis against climatic and WTD data (Table S7 in S1 File). We find that higher summer temperatures (June, July, August; JJA) are generally associated with higher aPAR values, a pattern evident in both contemporary and palaeo-climatic datasets. For the contemporary data, the Theil-Sen slope estimates indicate positive associations between summer temperature and aPAR (average aPAR: $\beta = 0.30$; maximum aPAR: $\beta = 0.33$; Table S8 in S1 File Fig 3). Spearman's rank correlation supports this trend, showing a weakly significant relationship between average aPAR and JJA temperature ($p = 0.0875$), and a more robust association for minimum aPAR ($p = 0.0123$; S2).

Using the palaeo-climate dataset, the relationship is even stronger, with Theil-Sen $\beta$ of 0.52 for average aPAR and 0.55 for maximum aPAR. Spearman's rank correlation confirms the significance of these associations (minimum

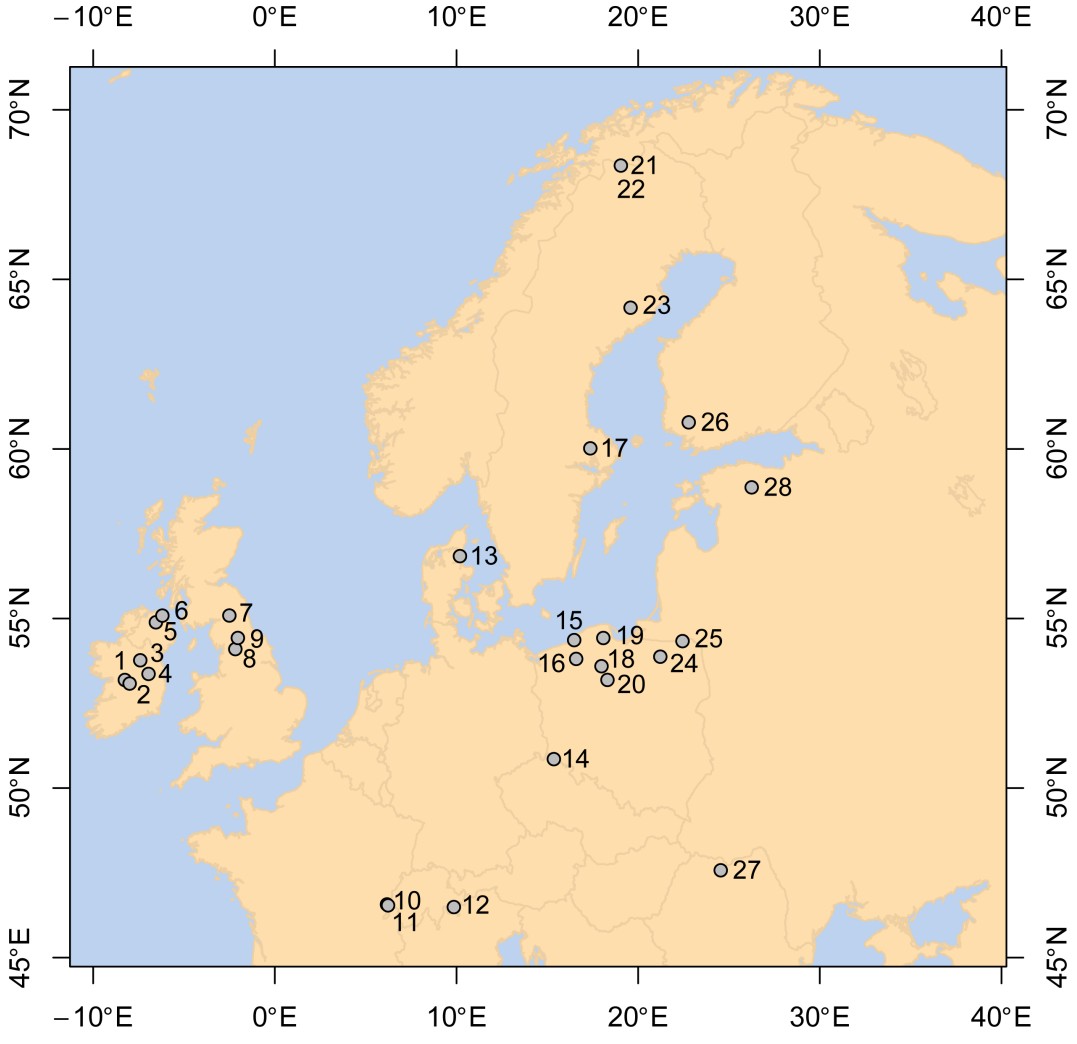

**Fig 1. Location of study sites.** Site descriptions are provided in Table 1.

aPAR: $p = 0.0101$; maximum aPAR: $p = 0.0025$; average aPAR: $p = 0.0004$). In contrast, relationships between aPAR and other climatic variables are ambiguous (Table S8 in S1 File, Fig 3, S2,5 & 6 in S1 File). We find that the highest aPAR values are found in the Scandinavia and Baltics region (S3) and that there is no discernible difference in aPAR between notable, continent-wide climate phases of the late Holocene, including the Little Ice Age (~1500–~1850 cal. yr. CE), Medieval Warm Period (~950–~1250 cal. yr. CE) and Roman Warm Period (~1–~400 cal. yr. CE) (S4).

We observe a weak negative relationship between average aPAR and average WTD (Theil-Sen beta = −0.31) (Fig 3). Spearman's rank correlation confirms a weakly significant relationship ($p = 0.0854$) (S2). aPAR is generally lower under deeper water tables (higher WTD), although a degree of non-linearity is suggested through the presence of the highest aPAR in the 5–10 cm WTD range (Fig 3). It is notable that none of our sites has high average aPAR when WTD is deeper than ~11 cm, suggesting a hydrological threshold. To explore this relationship in greater detail we re-examined the data from the 1-cm thick layers. Fig 4 shows there is a clear distribution of accumulation rates in relation to WTD – the highest box median values of AR occur when the reconstructed WTD is in the 5–10 cm range, which is in alignment with the site average-based analysis (Fig 3). Application of a Loess Smoothing Model indicates an optimal WTD range between 7 and

**Table 1. European study sites.**

| Site number | Site name | Region | Country | Latitude | Longitude | Reference |
|---|---|---|---|---|---|---|
| 1 | Cloonoolish | Britain & Ireland | Ireland | 53.1865 | −8.2569 | Blundell et al. (2008) Journal of Quaternary Science 23, 59–71. |
| 2 | Ballyduff | Britain & Ireland | Ireland | 53.0807 | −7.9925 | Swindles et al. (2013) Earth Science Reviews 126, 300–320. |
| 3 | Derragh | Britain & Ireland | Ireland | 53.7667 | −7.4083 | Langdon et al. (2012) Quaternary International 268, 145–155. |
| 4 | Ardkill | Britain & Ireland | Ireland | 53.3653 | −6.9532 | Blundell et al. (2008) Journal of Quaternary Science 23, 59–71. |
| 5 | Dead Island | Britain & Ireland | Ireland | 54.8862 | −6.5487 | Swindles et al. (2010) Quaternary Science Reviews 29, 1577–1589. |
| 6 | Slieveanorra | Britain & Ireland | Ireland | 55.0848 | −6.1921 | Swindles et al. (2010) Quaternary Science Reviews 29, 1577–1589. |
| 7 | Butterburn | Britain & Ireland | England | 55.0875 | −2.5036 | Mauquoy et al. (2008) Journal of Quaternary Science 23, 745–763. |
| 8 | Malham | Britain & Ireland | England | 54.0964 | −2.1750 | Turner et al. (2014) Quaternary Science Reviews 84, 65–85. |
| 9 | Keighley | Britain & Ireland | England | 54.4253 | −2.0369 | Blundell et al. (2016) Palaeogeography, Palaeoclimatology, Palaeoecology 443, 216–229. |
| 10 | Praz-Rodet | Continental Europe | Switzerland | 46.5667 | 6.1736 | Mitchell et al. (2001) The Holocene 11, 65–80. |
| 11 | Combe des Amburnex | Continental Europe | Switzerland | 46.5397 | 6.2317 | Sjögren & Lamentowicz (2008) Vegetation History and Archaeobotany 17, 185–197. |
| 12 | Mauntschas | Continental Europe | Switzerland | 46.4900 | 9.8544 | van der Knaap et al. (2011) Quaternary Science Reviews 30, 3467–3480. |
| 13 | Lille Vildmose | Scandinavia | Denmark | 56.8391 | 10.1896 | Mauquoy et al. (2008) Journal of Quaternary Science 23, 745–763. |
| 14 | Izery | Continental Europe | Poland | 50.8519 | 15.3602 | Kajukało et al. (2016) European Journal of Protistology 55, 165–180. |
| 15 | Słowińskie | Continental Europe | Poland | 54.3619 | 16.4785 | Lamentowicz et al. (2009) Boreas 38, 214–229. |
| 16 | Bagno Kusowo | Continental Europe | Poland | 53.8078 | 16.5872 | Lamentowicz et al. (2015) Palaeogeography, Palaeoclimatology, Palaeoecology 418, 261–277. |
| 17 | Åkerlänna Römosse | Scandinavia | Sweden | 60.0167 | 17.3667 | van der Linden et al. (2008) Palaeogeography, Palaeoclimatology, Palaeoecology 262, 1–31. |
| 18 | Jelenia Wyspa | Continental Europe | Poland | 53.5918 | 17.9821 | Lamentowicz et al. (2007) The Holocene 17, 1185–1196. |
| 19 | Stążki | Continental Europe | Poland | 54.4244 | 18.0833 | Lamentowicz et al. (2011) Studia Quaternaria 28, 3–16. |
| 20 | Linje | Continental Europe | Poland | 53.1880 | 18.3098 | Marcisz et al. (2015) Quaternary Science Reviews 112, 138–152. |
| 21 | Stordalen I | Scandinavia | Sweden | 68.3568 | 19.0484 | Gałka et al. (2017) Permafrost and Periglacial Processes 28, 589–604. |
| 22 | Stordalen II | Scandinavia | Sweden | 68.3568 | 19.0484 | Gałka et al. (2017) Permafrost and Periglacial Processes 28, 589–604. |
| 23 | Lappmyran | Scandinavia | Sweden | 64.1647 | 19.5828 | van der Linden et al. (2008) Palaeogeography, Palaeoclimatology, Palaeoecology 258, 1–27. |
| 24 | Gązwa | Continental Europe | Poland | 53.8726 | 21.2201 | Gałka et al. (2015) The Holocene 25, 421–434. |
| 25 | Mechacz | Continental Europe | Poland | 54.3314 | 22.4419 | Gałka et al. (2017) Quaternary Science Reviews 156, 90–106. |
| 26 | Kontolanrahka | NE Europe | Finland | 60.7833 | 22.7833 | Väliranta et al. (2007) The Holocene 17, 1093–1107. |

*(Continued)*

**Table 1.** (Continued)

| Site number | Site name | Region | Country | Latitude | Longitude | Reference |
|---|---|---|---|---|---|---|
| 27 | Tăul Muced | Continental Europe | Romania | 47.5739 | 24.5450 | Feurdean et al. (2015) The Holocene 25, 1179–1192. |
| 28 | Männikjärve | NE Europe | Estonia | 58.8667 | 26.2500 | Väliranta et al. (2012) Quaternary International 268, 34–43. |

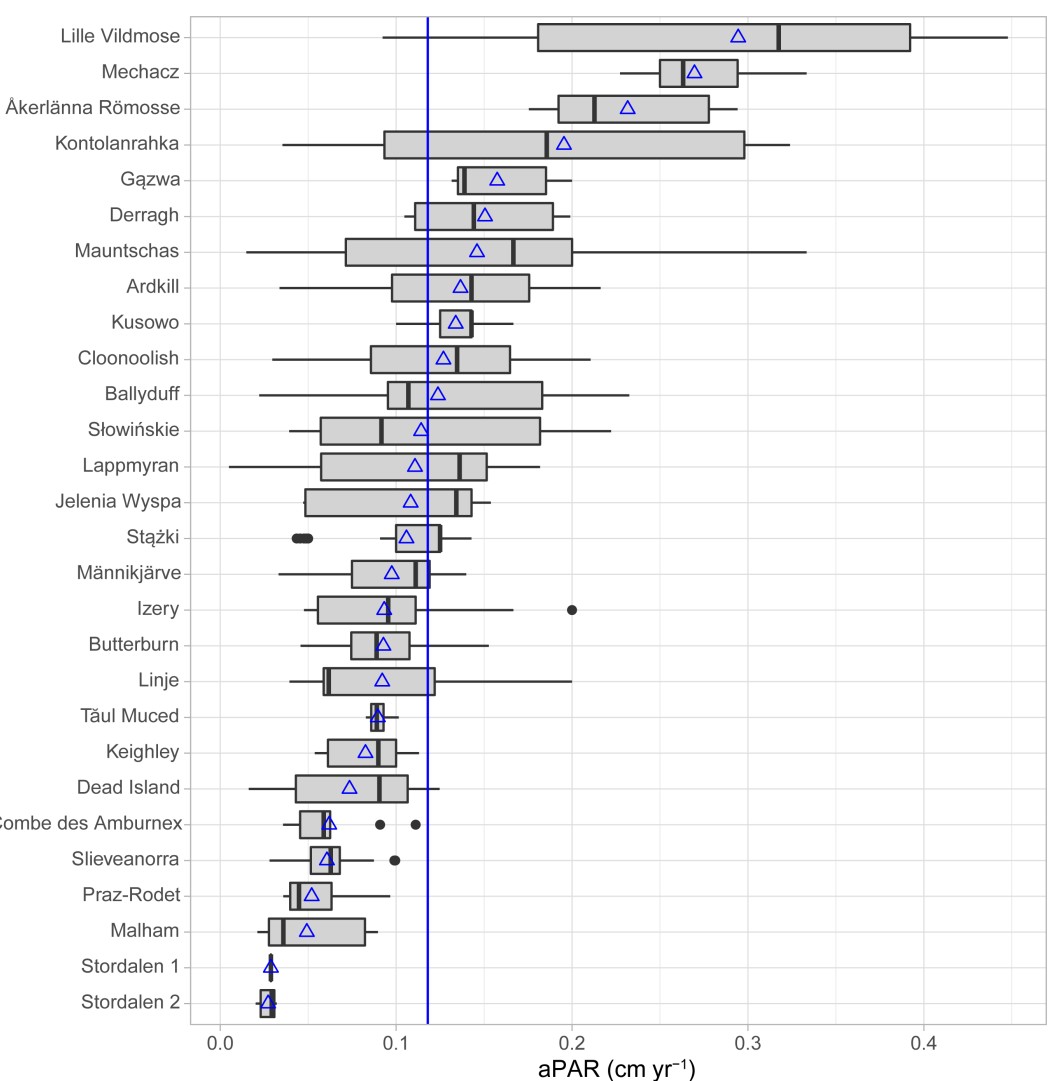

**Fig 2. Boxplot showing aPAR for each site.** Blue triangles indicate the mean values for each site. The blue line indicates the average of site means (0.118 cm yr⁻¹).

9 cm. Fitting a Gaussian Response Curve and Generalized Additive Model to the data suggests that the optimal WTD for the highest aPAR is 10.0 cm for both models (Fig 5). However, the Generalized Additive Model was not statistically significant, and it is worth noting that the estimated optimal water table depth (WTD) is close to the overall mean of the WTD data (10.4 cm). aPAR is generally low when water table depths are < 0 cm (standing water) or > 25 cm.

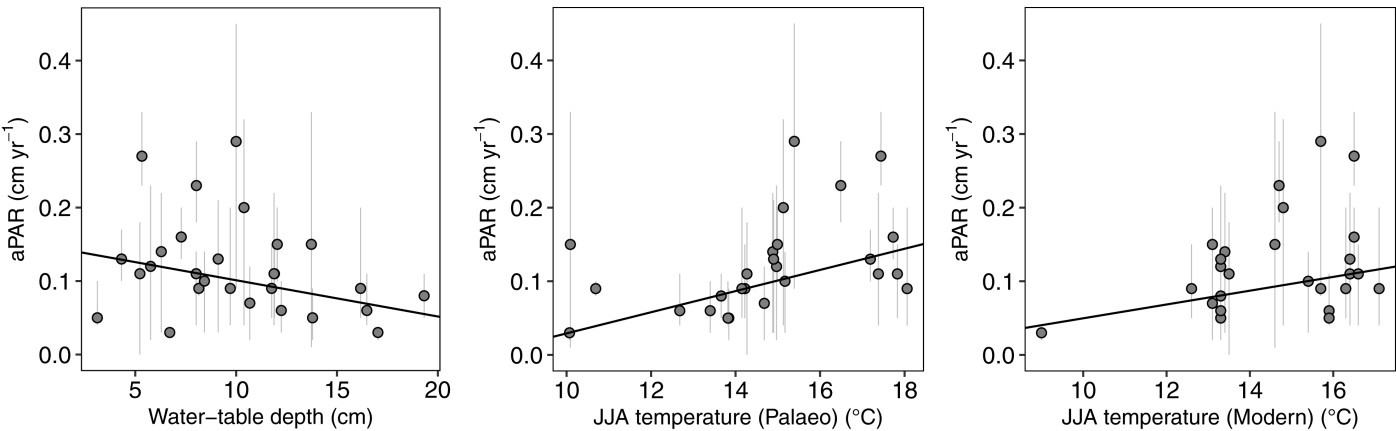

**Fig 3. Theil–Sen robust regression scatterplot of aPAR versus climatic data and reconstructed water-table depth (WTD).** Palaeo-climatic data are derived from CHELSA-TraCE21k [63], while contemporary data are sourced from the NOAA–CIRES–DOE 20th Century Reanalysis Version 3 [61,62].

## Discussion

Peatlands exist because of an imbalance between plant productivity and decomposition. However, the mechanisms behind peatland growth, stability and degradation have been debated for many years [19,22]. The role of restoration and management in maximising peatland carbon sequestration and storage potential – to maintain and enable their contributions as nature-based climate solutions to the climate crisis – is increasingly recognised in public and policy narratives. However, to optimise the benefits and minimise undesirable or ineffective outcomes, the driving mechanisms of peatland C accumulation need to be better understood. It is known that peatlands can self-regulate to a steady state [e.g., 23], but there is a concern that the magnitude of recent climate change and land-use degradation can lead to peatlands being pushed beyond a threshold, leading to loss of their carbon sink function and ultimately their carbon stock [13]. Although our findings support the common assertion that peat accumulates at ~0.1 cm yr$^{-1}$, they also suggest that aPARs vary widely from ~0.005–0.448 cm yr$^{-1}$ within Europe.

We use contemporary and palaeo- climatic data to quantify the relative difference in climatic space between our sites. Significant relationships between peat accumulation rates and climate variables have been found in previous studies [24–26]. In our study, the sites experiencing warmer temperatures – in particular, warmer summers – are associated generally with greater peat accumulation rates (Fig 3 & Table S8 in S1 File). This finding aligns with previous research that also indicated a positive association between elevated summer temperatures and increased peat accumulation rates [27], albeit with variations in temporal resolution. In high- to mid-latitude peatlands with adequate moisture, productivity is sometimes regarded as a more dominant driver of carbon accumulation than decomposition [28,29]. Warmer temperatures are well known to increase plant productivity, resulting in a greater input of organic matter into peat formations [1,4,30]. While microbial activity may also rise with increasing temperatures, increased productivity commonly outpaces decomposition, resulting in a net increase in peat accumulation [31]. However, for net peat accumulation there must still be adequate moisture in the summer months or elevated temperatures can lead to negative effects on *Sphagnum* structure and moisture holding capacity, desiccation, and even loss of peat [11,32]. In terms of the Köppen climate classification, the three sites with the highest aPAR values (Lille Vildmose, Mechacz and Åkerlänna Römosse) are in the warm-summer humid continental climate (Dfb), whereas the two sites with the lowest aPAR values (Stordalen I and II) are in the subarctic classification with cool summers and year-round rainfall (Dfc) and dark winter months with minimal insolation.

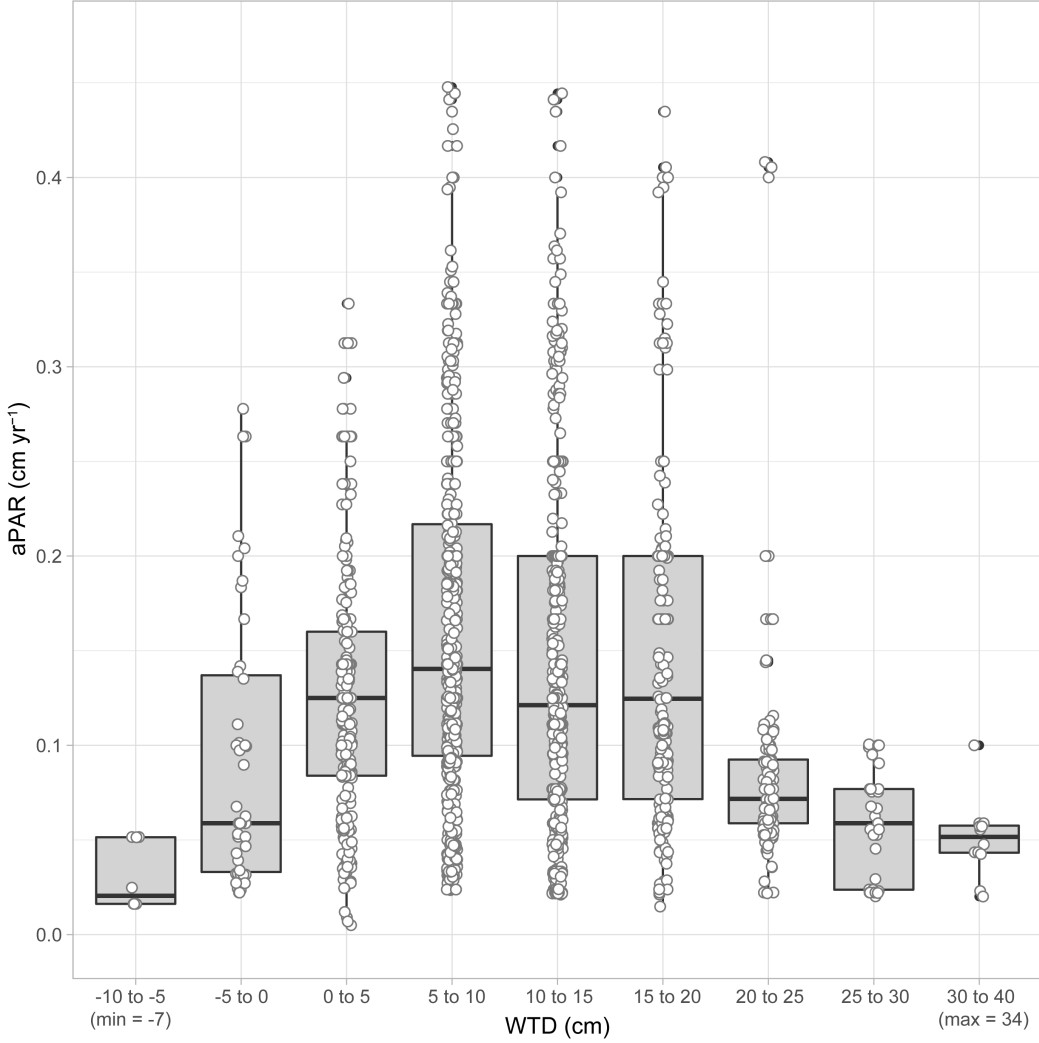

**Fig 4. Boxplot of aPAR separated into 5-cm bins across the water-table depth gradient.**

However, our findings suggest a lack of consistent or statistically significant correlations between peat accumulation and precipitation or GDD, as shown in Fig 3, S2 and S8 Tables in S1 File. While deglaciated regions have previously not shown a clear correlation with peat initiation and effective precipitation [e.g., 27], there may also be a degree of non-linearity, with an apparent optimum around 950 mm annual precipitation and 90 mm JJA precipitation (S5, S6). In such cases, overly wet conditions may reduce plant productivity by causing the expansion of open water areas within peatlands. This expansion can limit plant growth owing to reduced oxygen availability [11,33]. In waterlogged conditions, anaerobic decomposition processes dominate, which are much slower and less efficient, further favouring peat accumulation via decreased decomposition [33]. It is particularly intriguing in the case of GDD where previous studies [1,27,29] have intuitively indicated that a longer growing season leads to increased long-term carbon accumulation in northern peatlands. The apparent absence of a strong correlation between aPAR and GDD in this study may be explained by the need for more substantial variations in GDD values to establish a statistically significant relationship with peat accumulation

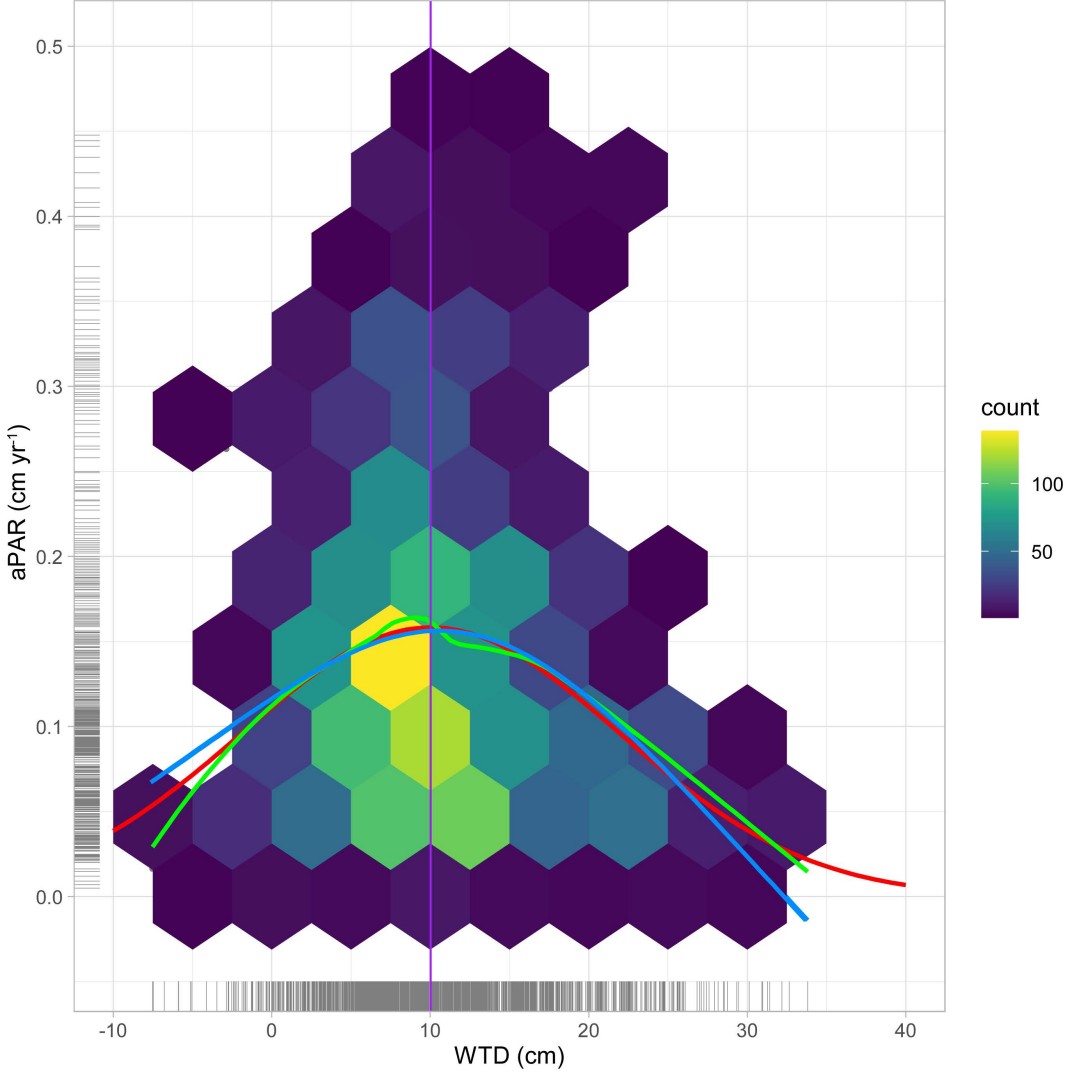

**Fig 5. Hexagonally-binned point density plots of aPAR data against mean water-table depth (width = 5 cm on the x-axis).** The Loess Smoothing Model (green), Gaussian Response Model (red) and Generalized Additive Model (blue) are illustrated. The dashed line represents a water-table depth of 10 cm. 'Rugs' are shown along the axes to illustrate data density.

rates. This limitation could be attributed to the relatively limited seasonal distinctions within our study areas, which might not encompass the full range of GDD values necessary to establish a connection with peat accumulation definitively.

Our research represents the first study using long-term peat accumulation data to examine the correlation between water-table depth and peat growth rates. We observe that the highest aPAR values occur when the water table is between 5 and 10 cm below the peatland surface. In contrast, aPAR tends to be low when the water table is above the surface (< 0 cm, ~standing water) or deeper than 25 cm, likely reflecting reduced plant productivity and increased decomposition losses, respectively. It is interesting that previous studies in peatland research have consistently identified an optimal WTD of approximately 10 cm, albeit with vastly contrasting approaches to this study, and over shorter timescales. Various studies focusing on the rewetting of peatlands suggest that maintaining a WTD of ~10 cm optimally mitigates greenhouse gas emissions. Studies include [34,35], drawing from earlier research [36–38], and [39], who refer to an 'optimal

rewetting scenario'. Furthermore, palaeoecological records from Polish peatlands suggest a plant community tipping point at approximately 11.7 cm WTD [40]. This WTD value represents a hydrological threshold where marked shifts in the dominant peat-forming species (vascular plants and mosses) occur, leading to changes in peat accumulation patterns and potentially increased carbon loss [40,41]. Indeed, increasing vascular plant cover usually promotes C loss by increasing heterotrophic respiration and decomposition of peat carbon owing to rhizosphere priming effects [41–43]. Independently of vascular plants, a drop in WTD can accelerate the growth and activity of microbial communities, causing a rise in heterotrophic respiration and decomposition rates [44]. [45] also demonstrated that peat formation rates peak when the acrotelm thickness is approximately 10 cm, while [46] found that photosynthesis reaches its maximum efficiency at a WTD of around 11 cm. A water-table depth value of 10 cm appears to also be an important hydrological threshold for the functional traits of testate amoebae [47].

Although [34,35,39] specifically mention an optimal peatland WTD of approximately 10 cm, their analyses are based on short-term observations, likely indicative of immediate peatland responses. Our approach provides information regarding the optimum WTD for long-term mean peat accumulation over millennia across multiple European sites. This approach provides valuable insights into the underlying process mechanisms that sustain the long-term stability of peatlands. While the accumulation rate may vary over time in individual sites due to factors such as climate and disturbances, our findings reveal a consistent regional pattern in European peatlands that has been sustained for millennia. This pattern highlights the presence of a common WTD that precedes optimal peatland growth – a phenomenon observed in both short-term experiments and our long-term palaeoenvironmental records. In other words, there is a numerical attractor in the peatlands where they tend to maintain a particular WTD that is conducive to the accumulation of peat and the overall stability of the ecosystem over millennia [e.g., 45, 48, 49].

The implications of our findings depend on the intended management objectives. If the aim is to maximise vertical peatland growth and thereby enhance carbon sequestration in the short term, the focus should be on evaluating whether previous peatland restoration programmes have adequately considered WTD position necessary to achieve those outcomes. In some cases, peatland restoration efforts may cause excessive waterlogging, which may be detrimental to peat accumulation and in turn lead to elevated $CH_4$ emissions [e.g., 50]. Thus, simply attempting to raise WTD without a scientifically informed understanding of peatland mechanisms may not yield the desired transformative results.

We acknowledge the potential limitations of our approach, especially regarding the determination of aPAR. We eliminated the potential problem of incomplete decomposition of the uppermost peats (the 'acrotelm effect') highlighted by [51] through removal of data from the period 1850 cal. yr. CE to present from our analysis, following [29]. However, a recent article has suggested instances where apparent carbon accumulation rates (aCAR) may contradict the true behaviour of the peatland [52]. The authors highlight the "ageing problem" – which stipulates that the use of aCAR from peatlands is erroneous because of slow, long-term decomposition of peat in the catotelm over millennia. This issue is also highly relevant for peat accumulation rates (aPAR) as discussed here, in that the thickness of the peat layer measured at the time of coring reflects its current state rather than what necessarily accumulated initially [52]. We note, however, that our peat core profiles for the past 2,000 years do not exhibit a long-term decay trend nor do they show statistically significant increases in bulk density downcore, suggesting that the ageing effect is unlikely to be a major problem in this study. Furthermore, [52] discuss how accumulation rates lower in the peat profile may have been affected by what happened later in time as water tables fluctuate (the 'secondary decomposition' problem). However, our use of a sufficiently large number of sites, and a single time period (the last two millennia), should allow the detection of a signal from the noise inherent to these systems (including a degree of secondary decomposition). Another potential criticism is the use of absolute reconstructed water-table measurements (cm) rather than directional shifts [53]. However, the transfer function used here has been shown to provide accurate mean annual water table predictions [54].

Despite these potential limitations, the consistent patterns observed in the accumulation rate data suggest that useful information may remain in peat archives. Our results may prove useful for parameterisation and testing of peatland development models such as Millennia [55], Holocene Peat Model [56] and Digibog [57]. Such models are becoming increasingly important for understanding the impacts of future climate- and land-use change on peatland ecosystems. Our results may support a growing body of research based on contemporary monitoring and palaeoecological investigation that the optimal WTD for maximum peat accumulation is ~10 cm in European peatlands, and to steer away from simply 'rewetting' without prior knowledge of optimal WTD for each specific system. This provides a better-informed target condition that can be used in the restoration process if the main priority is to encourage maximum peat accumulation and therefore carbon sequestration in the long term.

## Materials and methods

### Data processing

We analysed the European network of sites presented in [3] with each dataset spanning the last 2000 years (Table 1, Fig 1). A flexible Bayesian age–depth modelling approach [58] was used to generate an age model for each site (S1), using radiocarbon, $^{210}$Pb, tephra and spheroidal carbonaceous particle-based dating techniques. aPAR was calculated by dividing depth of peat accumulated (in cm) by time (years). [51] showed that recently formed peat cannot be compared to older, deeper peats as decomposition means that most of the newly added material will not become part of the long-term carbon store (the 'acrotelm effect'). We therefore removed aPAR data from the period AD1850-present from our analysis (following [29]) as this will contain uppermost peats where rapid aerobic decay is still taking place (the 'acrotelm' – [59]).

### Water-table reconstruction

WTDs were reconstructed from subfossil testate amoebae with the pan-European transfer function of [54] using a weighted averaging tolerance-downweighted model with inverse deshrinking (S9). Sample-specific errors of prediction (maximum and minimum reconstruction ranges) were generated through 1,000 bootstrap cycles. The pan-European transfer function model has been shown to generate accurate mean annual water table predictions for surface samples with associated automated instrumental mean annual and summer WTD measurements [54]. Therefore, we used the absolute reconstructed water-table values rather than standardised values in this study. Reconstructions were carried out on the subfossil testate amoeba datasets following the removal of weak silicic idiosomic tests (*Corythion-Trinema* type, *Euglypha ciliata* type and *Euglypha rotunda* type) [60].

### Climate analysis

Daily mean temperature and precipitation data were taken from the NOAA-CIRES-DOE 20th Century Reanalysis V3 dataset covering the period 1836–2015 [61,62]. The dataset is the first ensemble of sub-daily global atmospheric conditions spanning over a century, making it ideal for climate analyses extending as far back as the 19th century. We downloaded these data from KNMI Climate Explorer (https://climexp.knmi.nl/), focusing on the 1° longitude x 1° latitude grid box within which each of the 31 study sites is located. Growing degree days (GDD) were calculated for base temperatures above 0°C (GDD0) and 5°C (GDD5). Mean temperature, GDDs and precipitation totals were then aggregated to annual and northern hemisphere summer (June, July, August) temporal resolution for the entire record.

Mean monthly and annual palaeo-climate simulation data were taken from the CHELSA-TraCE21k long-term climatology covering the period 100 B.C.E. to 1990 C.E. [63]. The data are available at 100-year timesteps at high spatial resolution. We downloaded these data from: https://chelsa-climate.org/chelsa-trace21k/. Mean annual temperature (°C) and total annual precipitation (mm yr$^{-1}$) data were taken directly from the variables bio1 and bio12. Mean monthly temperature was calculated from the mean of tasmin and tasmax. Mean monthly temperature and precipitation totals were then aggregated for northern hemisphere summer (June, July, August) for each 100-yr timestep. For GDD0 and GDD5, we multiplied mean monthly temperatures by the days of each month and summed these values relative to thresholds of 0°C and 5°C respectively.

## Statistical analysis

Relationships between site-based aPAR (average, highest, lowest), climate variables and reconstructed WTD were explored for each site using Theil-Sen robust regression in the R package *deming* v.1.4 [64]. Theil-Sen regression coefficients were then standardised to produce beta coefficients. Spearman's rank correlation was used to further clarify relationships among variables because it is a non-parametric method that does not assume a linear relationship or normally distributed data. This makes it particularly suitable for exploring monotonic associations between ecological and climatic variables, especially when dealing with reconstructed or heterogeneous datasets that may include outliers or non-linear trends. The climatic variables analysed for each site included both contemporary (NOAA-CIRES-DOE 20th Century Reanalysis V3) and palaeo-climate (CHELSA-TraCE21k) datasets. These comprised annual and summer (JJA) temperature, annual and summer (JJA) precipitation, and growing degree days above 0°C (GDD0) and 5°C (GDD5). Water table depth (WTD) variables were represented by average, maximum, and minimum values from each reconstruction, based on bootstrapped error estimations.

To determine relationships between aPAR and WTD, a Loess Smoothing Model [65,66], a Gaussian Response Model [67], and a Generalized Additive Model (GAM) [68] were utilised. The GRM calculates an initial estimation of optimum and tolerance based on the weighted average, followed by a nonlinear optimization by the Levenberg-Marquardt method. All statistical analyses were carried out in either PAST v.4 [69] or R v.4.2.1 [70]. Graphics were produced using the *ggplot2* R package v.3.4.4 [71]. The GAM was calculated using the *mgcv* R package v.1.9.0 [72].

## Supporting information

**S1 File.** Supporting Information 1 (Figure): Bayesian age-depth models. Supporting Information 2 (Table): Spearman's rank correlation information ($R_s$ (bottom left) and p-values (top right) are shown). Supporting Information 3 (Figure): Boxplot showing aPAR for each region (Britain and Ireland, Continental Europe; Scandinavia and Baltics).Supporting Information 4 (Figure): Boxplot showing aPAR across different climate phases including the Little Ice Age (LIA: 1500–1850 CE), Medieval Warm Period (MWP: 950–1250 CE) and Roman Warm Period (RWP: 1–400 CE). All data points are shows as well as the non-Little Ice Age data points (nLIA). Supporting Information 5 (Figure): Theil–Sen robust regression scatterplot of aPAR versus modern climatic data (NOAA–CIRES–DOE 20th Century Reanalysis Version 3). Supporting Information 6 (Figure): Theil–Sen robust regression scatterplot of aPAR versus palaeo-climatic data (CHELSA-TraCE21k). Supporting Information 7 (Table): Site-based data for aPAR, WTD, contemporary climate and palaeo-climate. Supporting Information 8 (Table): Theil–Sen robust regression information for aPAR regressions with WTD, contemporary climate, and palaeo-climate. Supporting Information 9 (Dataset). All data.
(ZIP)

## Acknowledgments

We thank the two reviewers for their constructive comments on an earlier version of this manuscript. JMGs contribution represents NRCan contribution number/Numéro de contribution de RNCan: 20230392. This paper is a contribution to the PAGES C-PEAT group.

## Author contributions

**Conceptualization:** Graeme Swindles, Richard E. Fewster, Mariusz Lamentowicz, Evelyn M. Keaveney, Michelle M. McKeown.

**Data curation:** Graeme Swindles, Angela Gallego-Sala, Mariusz Lamentowicz, Matthew J. Amesbury, Antony Blundell, Frank M. Chambers, Dan J. Charman, Angelica Feurdean, Mariusz Galka, Edgar Karofeld, Atte Korhola, Lukasz Lamentowicz, Peter Langdon, Dmitri Mauquoy, Edward A. D. Mitchell, Gill Plunkett, Helen M. Roe, T. Edward Turner, Ulle Sillasoo, Minna Valiranta, Marjolein van der Linden, Barry Warner.

**Formal analysis:** Graeme Swindles, Donal J. Mullan, Neil T. Brannigan, Thomas G. Sim, Angela Gallego-Sala, Vincent E.J. Jassey.

**Investigation:** Graeme Swindles, Donal J. Mullan, Neil T. Brannigan, Richard E. Fewster, Thomas G. Sim, Maarten Blaauw, Mariusz Lamentowicz, Vincent E.J. Jassey, Katarzyna Marcisz, Sophie M. Green, Thomas P. Roland, Julie Loisel, Jennifer M. Galloway.

**Methodology:** Graeme Swindles, Donal J. Mullan, Neil T. Brannigan, Richard E. Fewster, Thomas G. Sim, Angela Gallego-Sala, Callum R.C. Evans.

**Project administration:** Graeme Swindles.

**Resources:** Graeme Swindles.

**Supervision:** Graeme Swindles.

**Validation:** Graeme Swindles.

**Visualization:** Graeme Swindles, Donal J. Mullan, Neil T. Brannigan.

**Writing – original draft:** Graeme Swindles, Donal J. Mullan, Neil T. Brannigan, Richard E. Fewster, Thomas G. Sim, Angela Gallego-Sala, Maarten Blaauw, Mariusz Lamentowicz, Vincent E.J. Jassey, Katarzyna Marcisz, Sophie M. Green, Thomas P. Roland, Julie Loisel, Callum R.C. Evans, Jennifer M. Galloway, Michelle M. McKeown.

**Writing – review & editing:** Graeme Swindles, Neil T. Brannigan, Richard E. Fewster, Thomas G. Sim, Maarten Blaauw, Mariusz Lamentowicz, Katarzyna Marcisz, Julie Loisel, Matthew J. Amesbury, Antony Blundell, Frank M. Chambers, Dan J. Charman, Angelica Feurdean, Jennifer M. Galloway, Mariusz Galka, Edgar Karofeld, Evelyn M. Keaveney, Atte Korhola, Lukasz Lamentowicz, Peter Langdon, Dmitri Mauquoy, Michelle M. McKeown, Edward A. D. Mitchell, Gill Plunkett, Helen M. Roe, T. Edward Turner, Ulle Sillasoo, Minna Valiranta, Marjolein van der Linden, Barry Warner.

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
