## [Decision Letter · Decision Letter 0]

14 Jan 2025

PONE-D-24-48416Do climatic or hydrological factors influence peat accumulation rates across Europe?PLOS ONE?

Dear Dr. Swindles,

Thank you for submitting your manuscript to PLOS ONE. After careful consideration, we feel that it has merit but does not fully meet PLOS ONE’s publication criteria as it currently stands. Therefore, we invite you to submit a revised version of the manuscript that addresses the points raised during the review process.

**ACADEMIC EDITOR:**
**Thanks very much for your submission! Two referees have submitted reviews. Although one reviewer suggests major revision (Reviewer 2) - and I agree that their questions regarding the relationship between aPAR and climate are important (i.e., major) - the revisions could be minor if answers are readily available. It is noteworthy that both referees recommended highlighting this paper on the general website.**

We look forward to receiving your revised manuscript.

Kind regards,

John Toland Van Stan II, Ph.D.

Academic Editor

PLOS ONE

Journal Requirements:

3. Please include your tables as part of your main manuscript and remove the individual files. Please note that supplementary tables (should remain/ be uploaded) as separate "supporting information" files.

Reviewers' comments:

Reviewer's Responses to Questions

**Comments to the Author**

1. Is the manuscript technically sound, and do the data support the conclusions?

Reviewer #1: Yes

Reviewer #2: Partly

2. Has the statistical analysis been performed appropriately and rigorously?

Reviewer #1: Yes

Reviewer #2: Yes

3. Have the authors made all data underlying the findings in their manuscript fully available?

Reviewer #1: Yes

Reviewer #2: No

4. Is the manuscript presented in an intelligible fashion and written in standard English?

Reviewer #1: Yes

Reviewer #2: Yes

Reviewer #1: This is an interesting manuscript that provides significant insights into peat accumulation rates across 28 well-dates European mires that are at least 2000 years old. The findings are particularly important for peat restoration efforts, particularly in the face of current and future climate change, suggesting that peat accumulation rates are controlled by complex interactions between climate, vegetation type and water depth, as well as finding a degree of geographic variation (potentially linked to these factors) across the study sites. A key finding is that 10 cm water table depths is crucial for peat restoration efforts and that further research is needed to underpin restoration efforts, particularly in terms of complexity of peat accumulation rates and how this can be applied to restoration efforts. Based on this I strongly support publication of this manuscript, as these findings will be of broad interest, not only to European researchers, but also other researchers across the globe as this provides an approach that will be directly relevant to other peatland systems, particularly in terms of underpinning restoration efforts of these vitally important ecosystems. There are a few minor issues (mainly related to referencing) that need to be addressed (listed below), and once these have been rectified the manuscript is ready for publication.

Line 159 to 160 - Dommain et al., 2010 listed with a publication year of 2016 in references (lines 456 to 461) - please verify the correct publication year.

Line 230 - Turetsky et al., 2010 listed with a publication year of 2015 in references (lines 584 to 585) - please verify the correct publication year.

Line 237 - Dorrepaal et al., 2003 listed with a publication year of 2004 in references (lines 462 to 464) - please verify the correct publication year.

Reviewer #2: Review of Swindles et al. PONE-D-24-48416 “Do climatic or hydrological factors influence peat accumulation rates across Europe?” Plos One.

Manuscript status

This manuscript is the first submission.

Comments

The manuscript aimed “to determine the climatic and hydrological controls on the apparent rates of peat accumulation (aPAR)” using “palaeoenvironmental data from 28 well-dated, intact European peat bogs”. The aPAR was analysed over the last 2000 years. The main finding is that the aPAR ranged between 0.005 and 0.448 cm yr-1. Although weak, there was a significant relationship between aPAR and water table depth (WTD) with an optimal aPAR at a WTD of -10 cm, which confirms the common result observed on shorter time series than in this study.

The study is impressive in gathering data from 28 sites in Europe, this is solid. However, there is a fundamental issue in the methods to estimate the relationship between aPAR and climatic variables relative to the relationship between aPAR and WTD:

L341-359

When reading the M&M on the WTD and climate reconstruction, it is understood that climate analysis was reconstructed for the period 1836-2015 while WTD reconstruction was done for the full peatland life durations with the subfossil testate amoebae. The reconstruction periods are different for both climate and WTD. It also means that climate does not have the same reconstruction range than WTD when examining the relationships with aPAR. Moreover, the M&M section shows that the climate reconstruction period 1836-2015 is almost not included in the aPAR period as the aPAR did not consider the peat accumulation since 1850AD (L339).

Is there something not explained in the M&M for the fitting between aPAR and climate? At first reading, it seems that the fitting is impossible as aPAR since 1850 is removed while climate data were available for 1836-2015. So, strictly reading the description indicates that the reconstruction could not be done between aPAR and climate as the time periods do not coincide. It seems impossible because as a large part of the work is on this relationship. Something is really unclear. Was there a climate reconstruction before 1836 to be fitted with the aPAR? Otherwise, what time period was concretely and clearly used in the relationship between aPAR and climate? This comment is major as a large part of the manuscript concerns the relationship between aPAR and climate.

There is another minor comment:

In L121-124, the information on the WTD threshold vs peat accumulation rates is well synthesised in the abstract but it does not appear as concise in the results in L194-207 where only results for the WTD of 5-11 cm is shown. The < 0 cm and > 25 cm thresholds are also not discussed or presented in the discussion.

Therefore, either change the text here in the abstract to reflect the results or add details in the results and discussion to indicate the relationship or threshold for the < 0 cm and > 25 cm effects. The minor issue here is why we read on thresholds of < 0 cm and > 25 cm in the abstract while it does not appear at all in the manuscript.

Apart from these two comments, the manuscript was well written and structured.

Note to the editor for question 3: it is not possible to answer as we do not see where data would be available in the web portal. I find that enough data are available through the supplementary information, no need for more.

**Do you want your identity to be public for this peer review?** For information about this choice, including consent withdrawal, please see our Privacy Policy

Reviewer #1: **Yes: ** Patrick Moss

Reviewer #2: No

---

## [Author Response · Author response to Decision Letter 1]

21 May 2025

Please see attached file. All comments from reviewers have been addressed.

---

## [Decision Letter · Decision Letter 1]

15 Jun 2025

Climate and water-table levels regulate peat accumulation rates across Europe

PONE-D-24-48416R1

Dear Dr. Swindles,

We’re pleased to inform you that your manuscript has been judged scientifically suitable for publication and will be formally accepted for publication once it meets all outstanding technical requirements.

Kind regards,

John Toland Van Stan II, Ph.D.

Academic Editor

PLOS ONE

Additional Editor Comments (optional):

Congratulations - your revised article is accepted. I have also reached out to PLOS One recommending that your work be featured. Thanks for submitting your excellent work.

Reviewers' comments:

Reviewer's Responses to Questions

**Comments to the Author**

Reviewer #1: All comments have been addressed

Reviewer #2: All comments have been addressed

2. Is the manuscript technically sound, and do the data support the conclusions?

Reviewer #1: Yes

Reviewer #2: Yes

3. Has the statistical analysis been performed appropriately and rigorously?

Reviewer #1: Yes

Reviewer #2: Yes

4. Have the authors made all data underlying the findings in their manuscript fully available?

Reviewer #1: Yes

Reviewer #2: Yes

5. Is the manuscript presented in an intelligible fashion and written in standard English?

Reviewer #1: Yes

Reviewer #2: Yes

Reviewer #1: All of my comments have been addressed, and the revised manuscript is very well structured, written and presented - providing important insight into the utilization of palaeoecological data that is highly relevant for peatland restoration - particularly in terms of the variable response of different systems to climate, which needs to be considered for restoration activities - which will be of broad interest.

Reviewer #2: The authors addressed all my comments, especially by adding the new climate analysis. I recommend the publication of this comprehensive work.

**Do you want your identity to be public for this peer review?** For information about this choice, including consent withdrawal, please see our Privacy Policy

Reviewer #1: **Yes: ** Patrick Moss

Reviewer #2: No

---

## [Editor Report · Acceptance letter]

PONE-D-24-48416R1

PLOS ONE

Dear Dr. Swindles,

I'm pleased to inform you that your manuscript has been deemed suitable for publication in PLOS ONE. Congratulations! Your manuscript is now being handed over to our production team.

Kind regards,

on behalf of

Dr. John Toland Van Stan II

Academic Editor

PLOS ONE